# Single-dose replicon RNA Sudan virus vaccine uniformly protects female guinea pigs from disease

Kyle L. O'Donnell [1], Hanna Anhalt [1], Greg Saturday [2], Nikole L. Warner [3], Troy Hinkley [3], E. Taylor Stone [3], Kiara Hatzakis [3], Amit P. Khandhar [3], Logan Banadyga [4,5], Jesse H. Erasmus [3] & Andrea Marzi [1] ✉

The Sudan virus (SUDV) outbreaks in Uganda in 2022 and 2025 created public health concerns in-country and the entire East African region. There are currently no licensed countermeasures against SUDV. We developed a SUDV vaccine candidate based on a nanocarrier (LION™) complexed with an alphavirus-based replicon RNA. Here, we compare the protective efficacy of the LION-SUDV vaccine either encoding the SUDV glycoprotein (GP) alone or in combination with the Ebola virus (EBOV) GP (LION-Combination). A LION-EBOV vaccine which is protective against EBOV was also included to determine the potential for cross-protection against SUDV infection. Single-dose vaccinations were conducted three weeks before challenge with a lethal dose of guinea pig-adapted SUDV using a female guinea pig disease model. We demonstrate 100% survival and protection with the LION-SUDV and the LION-Combination vaccines, while the LION-EBOV vaccine achieved 50% protection. Antigen-specific humoral responses correlate with decreased virus replication and survival. This result warrants further studies in larger animal species to ensure that protective efficacy is maintained with the single-dose LION-SUDV vaccine.

The 2022–2023 Sudan virus (SUDV) outbreak in Uganda has once again brought filoviruses to the forefront of public health concerns in the affected region as well as the global community[1]. After Ebola virus (EBOV) and Marburg virus, SUDV has been responsible for the most outbreaks of human filovirus disease, primarily occurring in Uganda. The largest outbreak of SUDV disease occurred in Gulu, Uganda, in 2000, causing 425 cases and 224 fatalities[2]. The 2022 SUDV outbreak started when a young man was diagnosed on September 19, 2022. Subsequent cases were reported in multiple districts in the central and western regions of Uganda, including Buyangabu, Kampala, Wasiko, Kagadi, Kyegegwa, Mubende, and Kassanda[3]. The outbreak was declared over by the Uganda Ministry of Health on January 11th, 2023,

with a cumulative confirmed case count of 142 and a 39% case fatality rate[3]. Currently, there is an outbreak ongoing in Uganda's capital, Kampala[4].

Filovirus countermeasure development progressed rapidly following the 2013–2016 West African EBOV epidemic. There are currently two licensed vaccines for EBOV, the single-dose vesicular stomatitis virus (VSV)-EBOV vaccine (Ervebo)[5] and the two-dose Ad26.ZEBOV/MVA-BN-Filo (Zabdeno/Mvabea) vaccine[6]. The VSV-EBOV vaccine provides limited to no cross-protection to SUDV due to antigenic differences in the glycoprotein (GP) used as the antigen[7,8]. The potential for Ad26.ZEBOV/MVA-BN-Filo to be cross-protective is greater since MVA-BN-Filo includes the SUDV GP as an antigen;

[1]Laboratory of Virology, Division of Intramural Research, National Institute of Allergy and Infectious Diseases, National Institutes of Health, Hamilton, MT, USA. [2]Rocky Mountain Veterinary Branch, Division of Intramural Research, National Institute of Allergy and Infectious Diseases, National Institutes of Health, Hamilton, MT, USA. [3]HDT Bio, Seattle, WA, USA. [4]Special Pathogens Program National Microbiology Laboratory, Public Health Agency of Canada, Winnipeg, MB, Canada. [5]Department of Medical Microbiology and Infectious Diseases, University of Manitoba, Winnipeg, MB, Canada. ✉e-mail: marzia@niaid.nih.gov

however, cross-protection has yet to be investigated in a clinical setting. Multiple other vaccine platforms are in preclinical development for SUDV, many of them with previous promising results for EBOV. These platforms include viral vectors, virus-like particles, subunit, and DNA vaccines in species-specific as well as pan-filovirus approaches[9]. A desirable product profile for an SUDV vaccine includes rapid protection after a single dose, due to the unpredictable and focal nature of outbreaks, where ring vaccination strategies have so far proven effective in the control of outbreaks while supporting licensure of experimental vaccines. So far, there has been limited development of an RNA vaccine for SUDV.

Two types of RNA vaccines are typically pursued, those consisting of conventional mRNA and those consisting of replicon-based RNA (repRNA). Lipid nanoparticle (LNP) delivery of conventional mRNA is a widely used vaccine platform. However, the rapid onset of protective immunity following a single dose as well as the durability of protective responses induced by this approach, is challenging. In contrast, repRNA includes a viral replicase that amplifies the mRNA encoding the antigen in the target cell. This method results in the expression of large amounts of protein from fewer copies of RNA over a longer period of time[10,11].

Recently, LNP delivery of repRNA resulted in dose-sparing efficacy and generated evidence of more durable responses in humans compared to the LNP/mRNA approach and holds promise in achieving single-dose efficacy[12,13]. However, the dose-limiting reactogenicity of repRNA/LNP in humans is a major challenge, as higher doses may be required to achieve rapid onset of protective immunity after only one dose[14]. For the delivery of repRNA, oil-in-water emulsion-based cationic nanocarriers, including LION, have been shown to increase antigen expression while limiting off-target proinflammatory responses, which have been observed when formulated with LNP[15]. In contrast to traditional LNPs that protect the RNA via encapsulation, LION nanoparticles protect RNA via electrostatically adsorbing the RNA molecules to the outer surface of the vehicle. The repRNA/LION platform has previously been demonstrated to elicit robust antigen-specific humoral and cellular responses, offering a balanced immune response for protection[16–19]. This platform is the basis of a COVID-19 vaccine that received emergency use authorization in India after demonstrating robust safety and immunogenicity in over 3000 participants enrolled in a phase 2/3 trial; the first such approval of a repRNA-based vaccine since the first description of the technology in 1989[20]. To this day, there are no licensed vaccines based on the original repRNA platform technology using a virus-replicon particle (VRP) for delivery, even though efficacy against EBOV was shown a decade ago[21]. While it is unclear why VRP-packaged repRNA has yet to achieve licensure, it is likely that manufacturing complexity is a contributing factor due to the requirement for mammalian cell-based production and downstream purification processes, as well as a non-zero risk of off-target production of replication-competent virus[22]. In contrast, the manufacturing of LION and repRNA does not require mammalian cell culture, thereby greatly simplifying production. Based on the recent achievements for COVID-19 and the ongoing clinical development of the repRNA/LION platform for Crimean Congo hemorrhagic fever virus[19,23,24], the repRNA/LION platform is well-positioned for its application against re-emerging infectious diseases like SVD.

Here, we generated repRNA vaccines expressing either the SUDV GP, EBOV GP, or a combination of both and evaluated their immunogenicity in comparison to the VSV platform, on which the only FDA-approved filovirus vaccine is based. As we have previously compared LION and LNP formulations for delivery of repRNA[15], we focused the present study on comparing the mRNA and VSV platforms, the latter the only one licensed by the FDA and EMA for single-dose prevention of a filovirus disease. Such a comparison has yet to be reported, particularly in the context of single-dose efficacy. Here, we assess and demonstrate 100% protective efficacy of a single dose of the LION-

SUDV vaccine delivered 21 days prior to challenge in the uniformly lethal SUDV guinea pig disease model.

## Results

### repRNA/LION vaccination protects from lethal SUDV infection

We first constructed repRNAs encoding either the wild-type SUDV-Gulu GP nucleotide sequence or a codon-optimized version. Following evaluation of SUDV GP-binding antibody responses in mice immunized with each candidate, we opted to move forward with repRNA encoding the wild-type SUDV GP, as no difference was observed in antibody response between the two candidates (Supplementary Fig. 1). Antigen expression for the selected three repRNA/LION vaccines was also confirmed in baby hamster kidney (BHK) cells by immunofluorescence analysis (IFA) and Western blot (Supplementary Fig. 2).

We vaccinated groups of guinea pigs ($n = 6$) 21 days prior to challenge with either the repRNA/LION construct expressing SUDV GP (LION-SUDV), a combination of the SUDV and EBOV GPs (LION-Combination), or EBOV GP alone (LION-EBOV) to determine if a single dose could protect guinea pigs from lethal SUDV infection. As vaccines based on VSV are in development for all filoviruses[25], the VSV-SUDV vaccine that had previously demonstrated uniform protective efficacy in the nonhuman primate (NHP) model[26] was chosen as the most logical positive control vaccine for our study. We found that 6/6 guinea pigs vaccinated with the LION-SUDV or the LION-Combination vaccines survived the challenge with guinea pig-adapted SUDV (GPA-SUDV), lacking signs of disease, while only 3/6 guinea pigs vaccinated with the LION-EBOV vaccine survived the infection. Interestingly, only 5/6 guinea pigs vaccinated with VSV-SUDV survived challenge, with a single guinea pig succumbing late during the acute disease course. All the control guinea pigs succumbed to disease by 9 days post-challenge (DPC) (Fig. 1A). The survival outcome correlated with changes in body weight, as guinea pigs vaccinated with the LION-SUDV or the LION-Combination vaccines had significantly less body weight loss compared to the control guinea pigs (Fig. 1B). An additional indicator of complete protection without disease is the lack of a fever in the guinea pigs vaccinated with LION-SUDV (Fig. 1C). This data indicates robust protective efficacy of the LION-SUDV and the LION-Combination vaccine after single-dose administration.

### RepRNA/LION vaccines inhibit SUDV replication in guinea pigs

We determined the amount of virus in the blood, liver, and spleen of guinea pigs at 6 DPC to further characterize the level of vaccine-mediated protection in this model. In line with the survival and body weight data, guinea pigs vaccinated with the LION-SUDV, the LION-Combination, or the VSV-SUDV vaccines showed the strongest efficacies, with significantly less SUDV RNA and infectious virus in the blood (Fig. 2A, B). The guinea pigs vaccinated with LION-EBOV exhibited a range of viremia from undetectable to similar levels as the control animals, correlating with the 50% survival data. However, this variability in viremia resulted in no significant differences when compared to the LION-EBOV and control groups. Liver and spleen were also investigated because they represent primary tissues of SUDV infection. We observed a complete lack of SUDV RNA and infectious virus in liver and spleen samples from guinea pigs vaccinated with the LION-SUDV or LION-Combination vaccines (Fig. 2A, B). Like the viremia, the LION-EBOV-vaccinated guinea pigs exhibited a wide range of SUDV RNA levels and infectious virus in both tissues, resulting in no significant differences when compared to the control group. In contrast to the viremia data, the VSV-SUDV guinea pigs had low levels of viral RNA in the spleen and liver, but no infectious virus was isolated from either tissue (Fig. 2A, B). The virology findings are reflected in the observed tissue pathology, as guinea pigs vaccinated with the LION-SUDV, LION-Combination, or VSV-SUDV vaccines presented with no or only mild pathology in the spleen and liver and limited to no SUDV antigen-specific immunoreactivity (Fig. 3A–L). Histopathological analysis

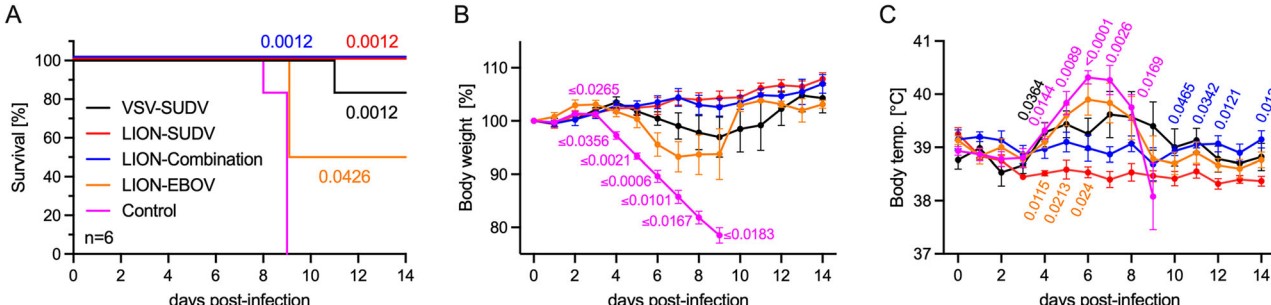

**Fig. 1 | Clinical data of vaccinated Guinea pigs challenged with GPA-SUDV. A** Survival, **B** body weight, and (**C**) body temperature changes of vaccinated and challenged guinea pigs ($n = 6$) through the acute disease phase of the study. Datasets display mean and standard error of the mean and were analyzed using Log-rank Mantel–Cox test (**A**) or two-way ANOVA/mixed-effects analysis with Dunn's multiple comparisons (**B**, **C**). Statistically significant differences are indicated incomparison to teh control group in (**A**) and for the control group in (**B**), with LION-EBOV only significantly different until 5 days post-infection. Significant temperature differences in comparison to the LION-SUDV group are indicated in (**C**).

revealed that the guinea pigs in the LION-EBOV and control groups had multifocal hepatocellular degeneration and necrosis with moderate neutrophilic infiltration and extensive vacuolar degeneration. Splenic pathology consisted of mild lymphoid depletion of the white pulp and multifocal red pulp necrosis with mild to moderate neutrophilic infiltrates and fibrin thrombi. A graphical representation of the histopathological scoring demonstrating differences in the liver and spleen is shown in Supplementary Fig. 3. In addition, liver and spleen samples in the LION-EBOV and control group showed high levels of SUDV antigen-specific immunoreactivity (Fig. 3M–T). This data highlights the superior protective efficacy of the LION-SUDV and LION-Combination vaccines compared to the LION-EBOV vaccine against GPA-SUDV challenge in guinea pigs.

**Vaccination elicits multifunctional humoral responses**

Quantitative analysis of the T cell response in guinea pigs is hampered by the limited availability of reagents. Therefore, we focused our analysis on characterizing the humoral immune response with the aim of identifying a correlate of protection. Serum samples from −7, 6, and 28 DPC were available for this analysis. At −7 DPC, guinea pigs vaccinated with VSV-SUDV, LION-SUDV, or LION-Combination developed robust antigen-specific IgG responses specific to the SUDV GP (Fig. 4A). At the same time point, only the LION-Combination and LION-EBOV-vaccinated guinea pigs had a specific IgG response to the EBOV GP (Fig. 4B), which was lower in magnitude compared to the response to the SUDV GP. At 6 DPC, the SUDV GP-specific IgG response was boosted in all groups; however, only guinea pigs vaccinated with the LION-SUDV- and LION-Combination vaccines developed a significantly higher response compared to the control guinea pigs, which succumbed to disease. At study end (28 DPC), there was no difference between any of the LION vaccine groups as surviving guinea pigs developed robust antigen-specific responses boosted by the GPA-SUDV infection. However, the VSV-SUDV vaccinated and surviving guinea pigs had significantly higher titers for both tested antigens compared to the LION-EBOV group. We investigated the antibody responses further by neutralization, antibody-dependent complement deposition (ADCD), and antibody-dependent cellular phagocytosis (ADCP). Consistent with the overall antigen-specific IgG titer, we found that, at −7 DPC, the guinea pigs vaccinated with VSV-SUDV and LION-SUDV presented with significantly higher neutralizing antibody titers compared to the control group and the guinea pigs vaccinated with LION-EBOV (Fig. 4C). At 6 DPC, the VSV-SUDV, LION-SUDV, and LION-Combination groups had significantly higher neutralization titers compared to the control group. At study end, no significant neutralization titer differences were detected between the groups (Fig. 4C). Finally, we characterized the Fc effector functions of this humoral response and discovered that the LION-SUDV and LION-

Combination groups had a propensity for stimulating ADCP and ADNP, while the VSV-SUDV-vaccinated guinea pigs elicited evidence of ADCD at 6 DPC (Fig. 4D–F). Overall, this data indicates that the VSV-SUDV, LION-SUDV, and LION-Combination vaccines are immunogenic and efficacious after single-dose administration against a lethal GPA-SUDV challenge. Platform-specific antibody functionality profiles were elicited, with the LION-SUDV vaccine stimulating a phagocytic and neutralizing functional profile, while the VSV-SUDV vaccine induced a neutralizing and complement-mediated functional profile.

## Discussion

The recent outbreak of SUDV disease in Uganda demonstrated that there is still a critical need for effective vaccines and therapeutics for this re-emerging highly pathogenic infectious disease. Despite the monumental efforts and successes in treating and vaccinating against EBOV disease, there is still a large gap in available countermeasures against other pathogenic filoviruses, including SUDV. While many vaccines have shown promising results in preclinical studies, only the VSV-SUDV and the chimpanzee adenoviral-vector have advanced to clinical trials[27–29].

Here, we demonstrated that uniform protection against lethal SUDV challenge can be achieved with the repRNA/LION vaccines in guinea pigs. All guinea pigs vaccinated with either a single dose of LION-SUDV or LION-Combination 21 days before challenge survived lethal infection with GPA-SUDV. We correlated the extent of the protective efficacy with decreased clinical signs, an absence of viremia and tissue viral loads, and a broader multifunctional humoral immune response. Interestingly, 50% of the guinea pigs vaccinated with the LION-EBOV vaccine were protected against SUDV challenge, despite the fact that these guinea pigs did not present with detectable cross-reactive antibodies to SUDV GP until 28 DPC. These results mirror findings from another vaccine study in which VSV-EBOV demonstrated 60% cross-protection in this guinea pig model[7]. Of note, in the same manuscript, the VSV-SUDV vaccine provided uniform protection, while in our study, there was 83% survival, with a single guinea pig succumbing late during the acute disease phase. The GP incorporated in our VSV-SUDV is based on the SUDV-Gulu variant, while the GP in the VSV-SUDV used in the manuscript by Cao and colleagues is based on the SUDV-Boneface variant, which is the homologous GP to the challenge virus, which is based on the SUDV-Boneface variant[7,30]. This mismatch did not impact protection mediated by the LION-SUDV vaccine, but it appears that with VSV-SUDV, this may matter in the outbred guinea pig model. This result could also simply be a feature of the outbred Harley guinea pig model used in our study. Whether or not the antigen mismatch would affect the immune responses in other outbred models like NHPs or efficacy in humans is unclear. For VSV-EBOV, at least in NHPs, the licensed vaccine expressing the EBOV-

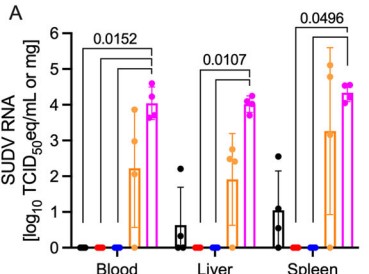
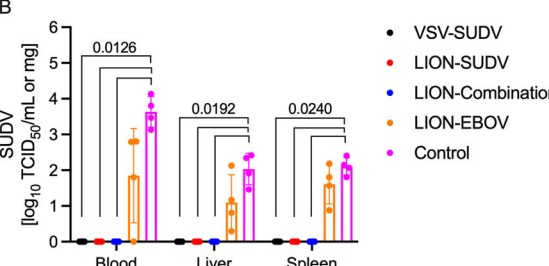

**Fig. 2 | Viremia and tissue viral loads in guinea pigs. A** SUDV RNA levels and (**B**) infectious SUDV titers in the blood, liver, and spleen of vaccinated and challenged guinea pigs (*n* = 4) collected 6 days post-infection. Datasets display geometric mean and geometric standard deviation and were analyzed using the Kruskal–Wallis test with Dunn's multiple comparisons. Statistically significant differences in comparison to the control group are indicated.

Kikwit GP is uniformly protective against EBOV-Makona challenge[31]. Similar observations have been made for MARV and Ravn virus (RAVV) with a VSV-MARV vaccine expressing the MARV-Musoke antigen demonstrating 100% protection in NHPs against challenge with MARV-Angola and RAVV[32].

A unique insight into vaccine platform differences with a focus on the humoral immune response was demonstrated in this study. While we did not measure significant quantitative changes, the quality of the humoral response was dependent on the vaccine platform. We found that VSV-SUDV induced a neutralizing response along with robust complement activation through ADCD. The role of complement activation in filovirus disease has been a double-edged sword, as there are indicators for both protective and non-protective contributions. Protective functions have been observed with antibodies targeting the glycan cap of the GP and demonstrated protective efficacy through complement c1q activation[33,34]. It has also been demonstrated that complement activation can lead to antibody-dependent enhancement (ADE) of EBOV infection through c1q activation in vitro[35]. The exact mechanism of ADE has not been fully elucidated; however, in vivo, it has been shown that NHPs treated with EBOV immune serum presented with a 100-fold increase in viral replication[36]. This phenomenon also occurs with monoclonal antibody treatment. A subset of non-neutralizing antibodies or neutralizing antibodies present at sub-neutralizing levels has been shown to correlate with increased viral replication at the primary site of infection[37,38]. This data highlights the importance of both the nature and magnitude of the antibody effector functions. In contrast, the repRNA/LION vaccines induced neutralization activity as well as high levels of ADCP and ADNP. It has previously been demonstrated that the long-term sustained immune response after EBOV infection in survivors is dependent on neutralizing antibodies and antibody-mediated phagocytosis; guinea pigs demonstrate the highest levels of antibody-mediated phagocytosis after LION-SUDV and LION-Combination vaccination, which may suggest the potential for increased protective efficacy with this vaccine platform[39,40]. We sought to further investigate whether overall antibody titers and antibody functionality types were true correlates of protection in this model. Implementing Spearman's correlation analysis, we determined that SUDV GP-specific IgG titers and ADNP were the earliest correlates with weight loss as a clinical sign at 4 DPC. Additional effector functions significantly correlated with ADCP and neutralization titers at 5 DPC, and the correlation increased as the disease progressed to 6 DPC (Supplementary Fig. 4; Supplementary Table 1). This correlation analysis revealed a significant negative correlation between peak temperature at 6 DPC and SUDV GP-specific IgG, ADCP, and ADNP, while no correlation with ADCD and EBOV GP-specific IgG was observed (Supplementary Fig. 4; Supplementary Table 1). We suspect that stimulating multiple mechanisms of phagocytosis targeting multiple cell types will make viral immune evasion more difficult.

A secondary goal of this study was to determine the cross-protection potential of the LION-EBOV vaccine against GPA-SUDV challenge. A similar study was conducted in the same guinea pig model with the VSV-EBOV vaccine, investigating whether VSV-EBOV could provide cross-protection to the SUDV challenge. The study concluded that 60% of the guinea pigs vaccinated with VSV-EBOV 28 days before lethal GPA-SUDV challenge were protected from disease and lethality[7]. Our results were similar, finding that 50% of the guinea pigs vaccinated with LION-EBOV were protected against GPA-SUDV. This was striking given that we measured lower overall antibody titers to EBOV GP and limited to no cross-reactive antibodies to SUDV GP until 28 DPC, which rose to surprisingly high levels at the conclusion of the study (Fig. 3A, B). We did measure moderate levels of neutralization on 6 DPC, but not to the level of significance. To give further insight into the utility of the repRNA platform, the same replicon system delivered by VRPs demonstrated complete protection against SUDV challenge in NHPs using a monovalent or bivalent vaccine[21]. Interestingly, cross-protection from EBOV back-challenge was only achieved using the bivalent vaccine but not the monovalent version[21].

A limitation of this study is the lack of investigation of the cellular immune responses elicited by the vaccines. It has been shown that repRNA/LION vaccines elicit a robust cellular response, but due to the limited immunological reagents available for guinea pigs, we were unable to investigate that aspect of the immune response and cannot exclude cross-reactive T cells as a potential mechanism of cross-protection[19]. Similar to the above-mentioned VSV study by Cao and colleagues, we would like to highlight that while the guinea pig model is one of the most stringent rodent models of disease for filovirus countermeasure development, it is likely less stringent than the NHP model, where cross-protection cannot be achieved as easily[7,41,42]. Additionally, this model uses an adapted virus, which may contribute to the differences in cross-protection that we demonstrate. The use of GPA-SUDV may be circumvented by using a model in which wild-type SUDV causes disease. Such a model now exists with the ferret, and future vaccine efficacy studies should be conducted in this model, allowing the assessment of protective efficacy after wild-type SUDV infection and similar disease progression[43]. Prior to clinical development, studies will need to be conducted in the more stringent NHP model with wild-type SUDV challenge before concrete claims of protection can be made. This would also address any limitations regarding the investigation of the cellular immune responses elicited by these vaccines. Future studies to further investigate the efficacy of the bivalent vaccine will need to be conducted to determine if the LION-Combination can also protect against EBOV challenge. Additionally, due to the large post-challenge boost of cross-reactive antibodies, it should be explored if there is potential cross-reactivity of the LION-SUDV to EBOV. Finally, additional studies are needed to solidify the protective efficacy of LION-EBOV against EBOV challenge.

In conclusion, a single administration of the LION-SUDV or LION-Combination vaccines uniformly protected guinea pigs against lethal

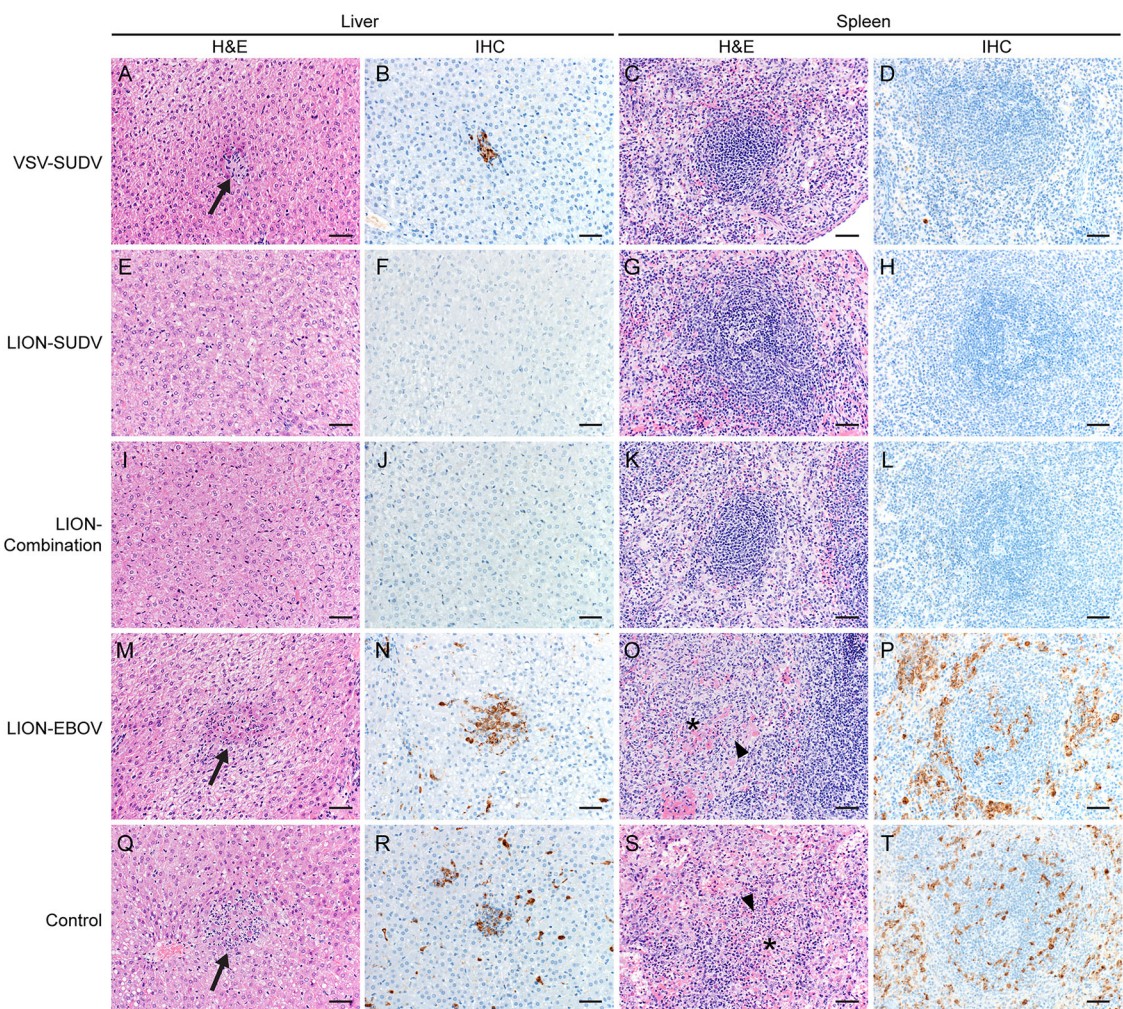

**Fig. 3 | Histopathology findings.** Histopathological analysis of liver and spleen samples of vaccinated and control guinea pigs collected at 6 days post-challenge (*n* = 4 per group). Depicted are hematoxylin & eosin (H&E)-stained representative tissue sections with corresponding immunohistochemistry (IHC) staining from one representative guinea pig per group. **A** Liver H&E, **B** liver IHC, **C** spleen H&E, and **D** spleen IHC of a VSV-SUDV vaccinated guinea pig. **E** Liver H&E, **F** liver IHC, **G** spleen H&E, and **H** spleen IHC of a LION-SUDV vaccinated guinea pig. **I** Liver H&E, **J** liver IHC, **K** spleen H&E, and **L** spleen IHC of a LION-Combination vaccinated guinea pig. **M** Liver H&E, **N** liver IHC, **O** spleen H&E, and **P** spleen IHC of a LION-EBOV vaccinated guinea pig. **Q** Liver H&E, **R** liver IHC, **S** spleen H&E, and **T** spleen IHC of a control guinea pig. Multifocal hepatocellular degeneration and necrosis (arrows) as well as splenic follicular lymphoid depletion (asterisks) with neutrophilic inflammation (arrowheads) were observed. 200× magnification; scale bar = 50 μm.

GPA-SUDV challenge. Both repRNA/LION vaccines demonstrated protection with no clinical manifestation, no viremia or measurable viral tissue deposition, and a multifaceted antigen-specific humoral response. Overall, both vaccines should be considered for additional animal studies to further characterize their preclinical efficacy and expand upon the immune correlates of protection each one elicits.

## Methods

### Ethics statement
All work involving infectious SUDV was performed following standard operating procedures (SOPs) approved by the Rocky Mountain Laboratories (RML) Institutional Biosafety Committee (IBC) in the maximum containment laboratory at the RML, Division of Intramural Research, National Institute of Allergy and Infectious Diseases, National Institutes of Health. Guinea pig work was performed in strict accordance with the recommendations described in the Guide for the Care and Use of Laboratory Animals of the National Institute of Health, the Office of Animal Welfare, and the United States Department of Agriculture and was approved by the RML Institutional Animal Care and Use Committee (IACUC). All mouse experiments were approved

by the University of Washington IACUC. Mice (*n* = 5 per cage) and guinea pigs (*n* = 2 per cage) were housed under controlled conditions of humidity, temperature, and light (12-h light:12-h dark cycles). Food and water were available *ad libitum*. Guinea pigs and mice were monitored and fed commercial chow.

### Cell and viruses
Vero E6 cells (*Mycoplasma* negative; ATCC, Cat. No. CRL-1586) and BHK-21 cells (*Mycoplasma* negative; ATCC, Cat. No. CCL-10) were grown in Dulbecco's modified eagle medium (DMEM) containing 10% FBS (Wisent Inc., St. Bruno, Canada), 2 mM L-glutamine, 50 U/mL penicillin, and 50 mg/mL streptomycin (all supplements from Thermo Fisher Scientific). THP-1 cells (*Mycoplasma* negative; ATCC, Cat. No. TIB-202) were grown at 37 °C and 5% $CO_2$ in Roswell Park Memorial Institute medium (RPMI) (Sigma-Aldrich, St. Louis, MO) containing 10% FBS, 2 mM L-glutamine, 50 U/mL penicillin, and 50 mg/mL streptomycin. HL-60 cells (ATCC, Cat. No. CCL-240) were propagated in Iscove's Modified Dulbecco's medium (IMDM) with 20% FBS, 2 mM L-glutamine, 50 U/mL penicillin, and 50 mg/mL streptomycin and differentiated with 1.3% DMSO for 5 days. The

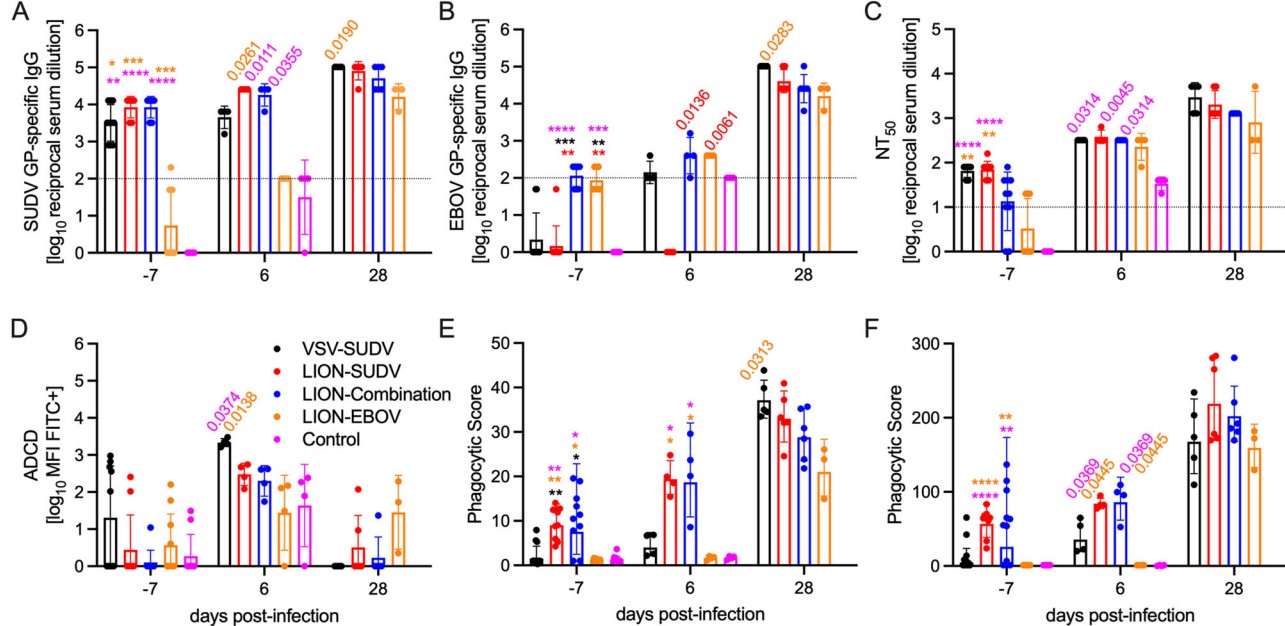

**Fig. 4 | Antigen-specific humoral responses and Fc effector function profiles.**
**A** SUDV GP-, **B** EBOV GP-specific IgG responses, and (**C**) neutralization activity in guinea pig serum. **D** Antibody-dependent complement deposition (ADCD), **E** antibody-dependent cellular phagocytosis (ADCP) activity, and (**F**) antibody-dependent neutrophil phagocytosis (ADNP) in serum. Datasets display geometric mean and geometric standard deviation and were analyzed using the Kruskal–Wallis test with Dunn's multiple comparisons. The dotted line represents the assay's limit of detection (assay starting dilution; negative samples were assigned the value 1). Statistically significant differences are indicated in colors corresponding to the vaccine group as follows: **A** day −7: *$p = 0.0379$, **$p = 0.0041$, ***$p = 0.0005$, ****$p < 0.0001$; **B** day −7: **$p < 0.0047$, ***$p < 0.0003$, ****$p < 0.0001$; **C** day −7: **$p = 0.0013$, ***$p = 0.0008$, ****$p < 0.0001$; **E** day −7: *$p < 0.0343$, **$p < 0.0081$; day 6: *$p = 0.0412$; **F** day −7: **$p < 0.0031$, ****$p < 0.0001$.

guinea pig-adapted SUDV variant Boneface (GPA-SUDV; Genbank accession number KT750754.1)[30] was received from the Public Health Agency of Canada under a material transfer agreement and used for the guinea pig challenge without further propagation.

## Vaccines

The alphavirus-derived repRNA was diluted in RNAse-free water, and in a separate tube, the LION stock was diluted with a combination of RNAse-free water, 40% sucrose, and 100 mM citrate. Subsequently, the repRNA was mixed 1:1 with the LION and incubated for 30 min at 23 °C to allow for the repRNA/LION complexes to form. We used the same lot of the LION complexed with the same lot of the repRNA encoding either the wild-type full-length GP of SUDV-Gulu or a codon-optimized version in the mouse study. The guinea pig vaccine study used only the repRNA constructs for the wild-type full-length GP of SUDV-Gulu and EBOV-Kikwit. The control cohort was vaccinated with a repRNA/LION vaccine encoding the Crimean Congo hemorrhagic fever virus nucleoprotein[19]. The VSV-SUDV was generated by cloning the SUDV-Gulu GP gene (GenBank NC_006432.1; 8A version) into the VSV backbone[26].

## Vaccine antigen expression

BHK-21 cells were transfected with in vitro transcribed repRNA at a dose of 32 ng/μL complexed with LION. Cells were allowed to incubate in repRNA/LION + OptiMEM reduced serum media for 30 min at 37 °C prior to the addition of complete media. Twenty-four hours after transfection, cells were harvested for IFA or Western blot analysis.

**IFA.** Cells were fixed, permeabilized, and stained with either an EBOV/ SUDV GP cross-reactive (α-zGP42/3.7, diluted 1:2000; a kind gift from Dr. A. Takada, Hokkaido University, Japan) or an EBOV GP-specific mouse monoclonal antibody (α-ZEBOV GP 12/1.1, 1:1000 dilution; a kind gift from Dr. A. Takada, Hokkaido University, Japan) and visualized with a goat α-mouse secondary antibody conjugated to

AlexaFluor594 (Thermo Fisher, Cat. No. A1105, diluted 1:500). Hoechst was used to visualize nuclei and images were taken at 100× magnification on a Keyence BZ-X Series All-in-one fluorescence microscope.

**Western blot.** Cells were harvested into RIPA buffer, and cell lysates were run in a non-reduced condition on Bolt™ Bis-Tris Plus Mini Protein Gels, 4%–12%, 1.0 mm, WedgeWell™ format (Thermo Fisher, Cat. No. NW04125BOX). Proteins bands were stained with either an EBOV/ SUDV GP cross-reactive (α-zGP42/3.7; 1:10,000 dilution) or an EBOV GP-specific mouse monoclonal antibody (α-EBOV GP 12/1.1; 1:5000 dilution) or a beta-actin antibody (GeneTex, Cat. No. GTX109639; 1:5000 dilution) and visualized with a goat α-mouse secondary antibody conjugated to horseradish peroxidase (Southern Bio, Cat. No. 4700-05; 1:10,000 dilution). All gels were visualized using Thermo Fisher Supersignal WestPicoPLUS Chemiluminescent Substrate (Cat. No. 34577) and imaged on a BioRad Gel Doc.

## Animal study design

Thirty-five female C57BL/6J mice (obtained from Jackson Laboratories), aged 6–8 weeks, were randomly assigned to 7 study groups, $n = 5$ for each vaccine dose. On day 0, mice were given an intramuscular injection of either 10 μg, 1 μg, or 0.1 μg, in a total volume of 50 μl of either wild-type SUDV GP or codon-optimized SUDV GP repRNA formulated with LION vaccine in the hind leg and were boosted on day 28. A control group was kept naïve. Blood samples were collected via retro-orbital or submental bleed on day 28. The mice were euthanized on day 42 for serum collection. Five mice were needed to assess differences in absorbance values of humoral immune responses assessed by power analysis using a two-sample $t$-test with an alpha value of 0.05 and a power of 0.8.

Fifty female Hartley guinea pigs (*Cavia porcellus*; obtained from Envigo), aged 6–8 weeks, were used for this study (based on the model being developed with only female guinea pigs[28]) and randomly assigned to 5 study groups, $n = 10$ for each vaccine. Guinea pigs were vaccinated

on −21 DPC with a single intramuscular injection of $1 \times 10^5$ plaque-forming units of VSV-SUDV or 20 µg of the respective repRNA/LION constructs. A blood sample was collected from the cranial *vena cava* on −7 DPC in ABSL2 instead of a blood sample collection on 0 DPC, as this procedure is complicated in ABSL4. All guinea pigs were challenged on 0 DPC with 1000 $LD_{50}$ (target 53 median tissue culture infectious doses ($TCID_{50}$), back titrated to 63 $TCID_{50}$). Daily temperatures and weights were taken until 14 DPC when all surviving animals returned to baseline. At 6 DPC, 4 guinea pigs per group were randomly chosen to be euthanized for sample collection to determine viremia, tissue viral loads, and antibody levels. The remaining guinea pigs ($n = 6$) were observed daily for clinical signs of disease and euthanized when IACUC-approved endpoint criteria were met. The study end was 28 DPC.

Power analysis with the assumption that 100% (99.9999%) of the animals in the control group and 25% of the animals in a vaccine group succumb to challenge would require group sizes of 6 to achieve a significant difference with 0.8 power and an alpha value of 0.05 using Fisher's exact test one-sided hypothesis. Four guinea pigs per necropsy group were required to assess differences in immune responses and virus loads appropriately, reaffirmed by one-way variance power analysis with an alpha value of 0.05, with a power of 0.8.

## Viral quantification

Levels of SUDV RNA in EDTA blood and tissue samples collected 6 DPC were determined using a RT-qPCR assay specific to the SUDV GP (forward primer CAAAGGGAAGAATCTCCGACC; reverse primer CAGGGGAATTCTTTGGAACC; probe GGCCACCAGGAAGTATTCGGACC). Blood samples were extracted with QIAmp Viral RNA Mini kit (Qiagen, Hilden, Germany), and tissue samples were processed using the RNeasy Mini kit (Qiagen) according to manufacturer specifications. One-step RT-qPCR was performed with QuantiFast Probe RT-PCR + ROX Vial Kit (Qiagen) on the Rotor-Gene Q (Qiagen). RNA from the GPA-SUDV stock was extracted in the same way and used alongside samples as standards with known $TCID_{50}$ concentrations. GPA-SUDV titers were determined using a $TCID_{50}$ assay as follows: confluent monolayers of Vero E6 cells in 48-well plates were inoculated in triplicate with 100 µL of serial 10-fold dilutions of the sample prepared in DMEM/2% FBS and incubated at 37 °C for 1 h. Following the 1-h incubation, 400 µL of DMEM/2% FBS was added to each well, and the cells were incubated at 37 °C for 14 days, at which time CPE was assessed. Titers were calculated using the Reed and Muench method[44].

## Histopathology

Necropsies and tissue sampling were performed according to IBC-approved SOPs. Tissues were cut into cassettes and fixed in 10% formalin for a minimum of 7 days. Tissues were processed with a Sakura VIP-6 TissTue Tek, on a 12 h automated schedule, using a graded series of ethanol, xylene, and ParaPlast Extra. Embedded tissues were sectioned at 5 µm and dried overnight at 42 °C prior to staining with hematoxylin and eosin. All tissue slides were evaluated by a board-certified veterinary pathologist. Pathology scores were determined following this scoring system: 0 = no lesions; 1 = small number of necrotic cells; 2 = moderate necrosis; 3 = significant necrosis; 4 = coalescing necrosis; 5 = diffuse necrosis.

## Enzyme-linked immunosorbent assays

The SUDV GP-specific IgG titers in mouse serum samples were determined at 1:100 dilution using ELISA kits following the manufacturer's instructions (Alpha Diagnostics, San Antonio, TX). Guinea pig ELISAs were conducted as follows: serum samples were inactivated by γ-irradiation and used in BSL2 according to IBC-approved SOPs. ELISA plates were coated with 1 µg/mL of recombinant SUDV GPΔTM or recombinant EBOV GPΔTM (IBT Bioservices, Rockville, MD, USA). The plates were washed three times with PBS/Tween and blocked with 5% milk buffer. Serum samples were diluted 1:50 for D-7 samples and 1:100

for D6 and D28 samples (limit of detection, as lower dilutions were not tested); 4-fold serial dilutions were generated thereafter to achieve measurable endpoint titers. A secondary antibody specific to guinea pig IgG was diluted 1:1000 (ab6908, Abcam, UK). The optical density at 405 nm was measured using a GloMAx Explorer baseline samples obtained from naïve guinea pig serum. The cutoff value was set as the mean OD plus three times the standard deviation from naïve guinea pig serum samples.

## Antibody effector function analysis

Assays to investigate antibody effector functions were adapted from established protocols[45] and are detailed below. Post-challenge guinea pig sera were inactivated by γ-irradiation (4 MRad) and removed from the maximum containment laboratory according to IBC-approved SOPs[46]. Recombinant SUDV GPΔTM (IBT Bioservices) was tethered to Fluospheres NutrAvidin-Microspheres yellow-green or red (Thermo Fisher Scientific, Waltham, MA) using the EZ-link Micro Sulfo-NHS-LC-Biotinylation kit (Thermo Fisher Scientific). Representative gating strategies are shown in Supplementary Fig. 4.

**ADCD.** Serum samples were heat-inactivated at 56 °C for 30 min then diluted 1:250 in DMEM and applied to the conjugated beads for 1 h at 37 °C. Next, guinea pig complement (Cedarlane, Burlington, Canada) was added for 30 min. A control plate was run in tandem with heat-inactivated guinea pig complement to account for the background signal. The bead complexes were washed with FACS buffer and stained with anti-C3c-FITC (Antibodies-Online). Data were acquired on a FACS Symphony (BD, Franklin Lakes, NJ) and analyzed in FlowJo v10.

**ADCP.** Serum samples were heat-inactivated at 56 °C for 30 min then diluted 1:250 in DMEM and applied to the conjugated beads for 1 h at 37 °C. The serum bead mixture was then transferred to a plate of THP-1 cells for 1 h at 37 °C. Data were acquired on a FACS Symphony (BD) and analyzed in FlowJo v10. Naïve serum samples were used to establish the baseline of non-specific phagocytosis. A phagocytic score was determined using the following formula: (percentage of FITC+ cells)*(geometric mean fluorescent intensity (gMFI) of the FITC+ cells)/10,000.

**ADNP.** Serum samples were diluted 1:250 in culture medium and incubated with GP-coated beads for 2 h at 37 °C. Beads (20 µl) were added to $5 \times 10^4$ cells/well HL-60 cells (differentiated with 1.3% DMSO to increase the expression of CD16) and incubated for 2 h at 37 °C. Cells were then stained for CD11b (Clone 3G8; BioLegend) and fixed with 4% paraformaldehyde. Data were acquired on a FACS Symphony (BD) and analyzed in FlowJo v10. Naïve serum samples were used to establish the baseline of non-specific phagocytosis. Neutrophils were defined as CD11b+. A phagocytic score was determined using the following formula: (percentage of FITC+ cells)*(geometric mean fluorescent intensity (gMFI) of the FITC+ cells)/10,000.

## Neutralization

Neutralizations were conducted with VSV-SUDV-GFP on Vero E6 cells seeded in 96-well plates. Serum samples were heat-inactivated at 56 °C for 30 min and 5-fold serially diluted in DMEM. VSV-SUDV-GFP was added in equal volumes at an MOI of 0.5, and the mixture was incubated for 1 h at 37 °C. The antibody-virus solution was then transferred to the cells and incubated for 18 h at 37 °C and 5% $CO_2$. Infected cells express GFP. The cells were fixed with 4% PFA and resuspended in FACs buffer. Naïve guinea pig serum samples were used to establish the baseline of the non-specific neutralization. Data were acquired on a FACS Symphony (BD) and analyzed in FlowJo v10.

## Statistical analysis

We assessed differences in the survival curves using log-rank analysis. All other data were obtained from distinct samples and compared

using a Kruskal–Wallis test with Dunn's multiple comparisons. Statistically significant differences are indicated with asterisks as follows: $p < 0.05$ (*), $p < 0.01$ (**), $p < 0.001$ (***), and $p < 0.0001$ (****). These statistical analyses were performed in Prism (version 9; GraphPad). Spearman correlation analyses between IgG endpoint titers and functionality readout and clinical symptoms were performed using the JMP® statistical analysis software. The $X–Y$ scatterplots show 95% confidence density ellipses for normally distributed data. The table indicates Spearman correlation coefficients that were statistically significant ($p < 0.05$).

## Reporting summary
Further information on research design is available in the Nature Portfolio Reporting Summary linked to this article.

## Data availability
Datasets generated and/or analyzed during the current study are appended as supplementary data. Source data are provided with this paper at https://doi.org/10.6084/m9.figshare.28771073.

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

## Acknowledgements

We thank members of the Rocky Mountain Veterinary Branch, NIAID, for supporting the guinea pig study and members of the Rocky Mountain Research Technology Branch, NIAID, for assistance with the histopathology figure generation. The project was funded by the Intramural Research Program, NIAID, NIH (AI001254 to A.M.). Funding was in part also provided by HDT Bio (internal funding to J.H.E.).

## Author contributions

K.L.O., J.H.E. and A.M. designed the studies. K.L.O., H.A., G.S., N.L.W., T.H., E.T.S., K.H., A.P.K., L.B. and A.M. performed the studies and carried out the analyses. J.H.E. and A.M. acquired funding for the study. A.M. supervised the study. K.L.O. and A.M. wrote the manuscript with input from all authors. All authors approved the content of the submitted manuscript.

## Funding

## Competing interests

N.L.W., E.T.S., K.H., T.H., A.P.K. and J.H.E. receive a salary and have equity interests in HDT Bio. J.H.E. is an inventor on US patent application no. 62/993,307 "Compositions and methods for delivery of RNA" pertaining to the LION formulation for vaccine development. There are no restrictions on the publication of research data. The remaining authors declare no competing interests.
