## [Transparent Peer Review file · Nature Communications]

Single-dose replicon RNA Sudan virus vaccine uniformly protects female guinea pigs from disease

Corresponding Author: Dr Andrea Marzi

Version 0:

Reviewer comments:

Reviewer #1

(Remarks to the Author)

The study introduces a vaccine platform utilizing replicon RNA (repRNA) complexed with LION™ nanocarriers. Key findings include that a single dose of the LION-SUDV encoding SUDV-GP and LION-Combination vaccines provided 100% protection against lethal SUDV challenge in guinea pigs, while the LION-EBOV vaccine encoding EBOV-GP achieved 50% cross-protection. The vaccines elicited robust and multifunctional antibody responses, correlating with decreased viral replication and clinical signs and enhanced survival rates.

Major Points:

The platform-specific differences in immune responses were noted when repRNA/LION vaccines were compared to VSV-based vaccines, with repRNA-LION stimulating a broader functional antibody profile compared to VSV. Generally, the data is well presented and demonstrate the efficacy of this vaccine platforms is associated with an increased phagocytotic activity profile of repRNA vaccine approach versus the VSV platform although both platforms are highly efficacious in this model. In addition, the absence of T-cell data, due to absence of standard method for Guinea is a significant limitation of study from the immunological perspective.

The novelty of this work mainly resides in the delivery method of the alphavirus RNA replicon, which uses LION as opposed to SUDV-GP virus-replicon particles (VRP). The latter was previously demonstrated to fully protect non-human primates from lethal SUDV challenge (<https://journals.asm.org/doi/epub/10.1128/jvi.03361-12>). Although technical details of the SUDV GP alphavirus repRNA construct are not clearly described in the methods section, they appear identical to those used by Herbert et al., as the authors later state in the discussion (lines 221 to 225). The main differences between this study and Herbert et al. 2013 lie in the use of guinea pigs versus non-human primates (NHPs) and the delivery method (LION vs VRP). To better highlight the advancement of this work over Herbert et al., the authors should expand demonstrate the advantages of LION over packaged repRNA in VRPs in term of efficacy and/or production. I think this paper would have greatly benefited from head-to-head comparison LION versus LNP and/or VRPs instead of VSV as we already knew that VRP delivered SUDV-GP repRNA was efficacious in NHPs. This would have demonstrated the potential of LION versus VRP delivery of the repRNA. To the best of knowledge, this type of comparative work has not been published for any repRNA vaccines.

Minor points:

Line 171-172 "This result was surprising, given that these guinea pigs did not exhibit detectable cross-reactive antibodies to SUDV GP until 28 DPC."

This statement is unexpected, as cross-reactive antibodies between Sudan virus and Ebola virus GPs have been described on multiple occasions. I believe the cutoff used for positivity might be responsible for this interpretation. In Figure 3A, B, C, the dotted line represents 'the assay limit of detection,' but the manuscript does not explain where this limit comes from. Additionally, upon reviewing the graphs, it is evident that several animals develop cross-reactive antibodies prior to the challenge, albeit at lower levels."

Reviewer #2

(Remarks to the Author)

This paper addresses an important topic, contributing to the response to the recent outbreak of Sudan virus. Having read through, I found several shortfalls which I have listed below.

MAJOR COMMENTS:

(i) The study in guinea pigs was concluded on day 14 post-challenge, whereas routinely day 21 (or even day 28) are widely used for these type of studies. From the clinical data (Figure 1), not all animals seem to be putting on weight and endpoints were reached on day 12. Therefore, the authors need to explain why the study was not scheduled to last longer to capture any animals with late-onset disease signs.

(ii) The histology analysis would greatly be aided if viral staining were performed instead of gross H&E changes. Could this staining be conducted? Given the importance of the histology findings, could these images be brought into the main manuscript?

(iii) Serum samples were collected at day -7, 6 and 28 DPC (line 132). I am not sure why samples weren't collected on the day of challenge, to provide important information on the immune results at the time of challenge which are perhaps the most useful.

(iv) For the statistical analysis, could the authors please double-check? In Fig 3C (described in line 146) it would appear that the LION-Combination group is significant values are all higher than the control group. Similar in line 147, the LION-EBOV group looks significantly higher but is not mentioned in the text or figure.

(v) Line 151-152 mentions about the VSV-SUDV vaccinated animals having evidence of ADCD. At the day -7 timepoint this was not significant so I can't see how this could be assigned to a vaccine response.

(vi) Line 154-156: In this section, I think that only the LION-SUDV and VSV-SUDV vaccines are directly comparable, as these are the only groups with the same antigen on different backgrounds. Therefore the argument about VSV giving a more robust profile is questionable.

(vii) Lines 170-172: The authors don't seem to correlate the 50% vaccinated with LION-EBOV being protected being identical to the 50% of animals in this group having anti-SUDV antibodies in Fig. 3A. It would be good to clarify if the animals having specific IgG were the ones which were provided protection against virus challenge.

(viii) Line 178-182: With the use of outbred animal species, outliers are often experienced. There is too much emphasis on this single animal and no consideration of effect being due to the nature of the biological systems used.

(ix) Line 193: Upon looking at ref 29, the authors appear to be selectively using data which support their work and arguments. The 100-fold increase is only observed in this reference at the day 7 timepoint, and for earlier timepoints viremia is actually lower.

(x) Lines 235-239: There are pros and cons with all animal model systems. Stating "...makes ferrets more favorable than the guinea pigs" is misleading. Whilst non-adapted virus is preferable, the infectious doses and disease progression kinetics could be argued that guinea pigs are the better system. The authors need to provide balanced arguments. Similarly, the argument of immunological reagents is invalid. Along with ferret-specific, there are also increasing guinea-pig specific reagents available. For T-cell analysis, there are mAb clones for IFN-gamma commercially available (e.g. MabTech) which can be used for flow cytometry, ELISA and ELISPOT assays.

(xi) The methods are very brief and miss crucial information. Whilst some details are in the supplementary documentation, key points are still absent. The source of the non-ATCC cell lines are required, especially to be assured of their authenticity. The differentiation of HL-60 cells to what state is not mentioned, nor assurance of the phenotype of the differentiated cells. The source of the guinea pig adapted virus is needed, and whether this strain is available by others, alongside the passage history during propagation. For the TCID50 method (line 307), key information on the analysis is missing, e.g. CPE assessment/crystal violet staining, controls to standardise outputs, etc. In histopathology (line 313), it just says "prior to staining" with no details on stains. For the ELISA (line 322), no information on the secondary antibody is provided. The antibody-dependent assays all lack antibody concentration details and controls/standards for interpretation. In the neutralisation assay (lines 353-359), I am unsure how the readout is on FACS - are cells stained to assess infectivity?

(xii) Lines 285-298: In the animal study design, groups of 4, 5 and 6 animals are used for different analysis. The reasoning for these numbers, e.g. power analysis, is critical.

MINOR COMMENTS:

Line 36: comma needed after '2000'.

Line 36: is there a better reference covering the outbreak in Uganda to reference 2?

Line 45: Reference 7 is cited, but key information on the blended vaccine, including a SUDV component, did provide protection. Appreciate that it was just assessed in n=2 animals though.

Line 47: add ';' after 'however'.

Line 60: space before 'Recently' needs adding.

Line 64: change to "Oil-in water...have been shown..."

Line 138: The significantly higher responses are mentioned, but what about the importance of the lower responses?

Line 169: "robust" is used to describe responses in contrast to "broad" in line 155. Needs consistency.

Lines 202-205: this belongs in the results section.

Lines 205-207: the "Additional effector functions" should be listed.

Line 206: The "5 DPC" is mentioned, but this doesn't fit with timepoints for the parameters in Fig 3.

Line 233: As ref 6 is cited earlier in sentence would remove from later in sentence.

Line 263: Mentioned mouse studies being approved by the University of Washington IACUC, but no authors are from this institute.

Line 302-303: Using stock TCID50 to quantify RT-PCR outputs is not optimal. Transcript control material for measuring genome copies is more relevant as there is variation in live virus quantification and viral RNA levels.

Fig 3E and F: Having ADCP and ADNP, respectively, on the x-axis similar to Fig 3D would be beneficial.

Reviewer #3

(Remarks to the Author)

Donnell et al evaluated the protective efficacy of candidate vaccines based on repRNA vaccine platform. One candidate expresses SUDV-Gulu GP nucleotide sequence (LION SUDV), the second one expresses SUDV and EBOV GPs (LION-Combination) and the third one expresses EBOV GP alone (LION-EBOV). These candidate vaccines were compared to VSV-SUDV vaccine (SUDV Boneface variant) in guinea pig model. The authors observed that a single dose of LION SUDV and LION-Combination induced 100% protection after a challenge with a guinea pig adapted-SUDV Boneface strain three weeks post-immunisation. Interestingly, LION-EBOV vaccination led to 50% protection against a SUDV challenge which means a cross-protection can be observed in some animals. Strong protection induced by LION SUDV and LION-combination correlated with low pathology, low viremia and viral loads in blood, liver and spleen. Protection also correlated with early functional antibody responses.

It is a very elegant and well-designed study. The paper is well written and the figures are quite clear.

Major comments:

Even though this repRNA vaccine platform has been previously described with other sequences, data linked to the LION SUDV and LION-combination vaccine design and formulation would be very relevant before showing immunogenicity and protection data. I mean a few data similar to Fig 1 (Erasmus et al 2020, <https://www.science.org/doi/10.1126/scitranslmed.abc9396>).

Line 121: As a histopathological analysis was performed by a pathologist, could the authors add a scoring system to their analysis? The scores could be used in the correlation analyses.

Suppl Fig 2: please add a scale to each picture. The pathology which is described in the text could be shown on the pictures using arrows.

Suppl Fig 4: Could the authors show representative plots for each experimental group?

Minor comment:

Please check the word Spearman in the manuscript. Sometimes it is written Spearman.

Version 2:

Reviewer comments:

Reviewer #2

(Remarks to the Author)

The authors have done a great job revising their manuscript, and this second version is a lot easier to read and interpret. I just had a couple of minor comments:

(1) For several of the rebuttals, the information and arguments are acceptable and justified (e.g. feasibility of working in the ABSL4 laboratory and restrictions with taking bleeds). However, as these haven't all been incorporated into the manuscript, the reader won't be privy to the information and reasonings. Therefore, incorporation of these reasonings into the manuscript would prevent the reader also having the same questions and concerns.

(2) The figure legends in the text are out of synchronisation with the incorporation of the histology into the main file, so need changing.

Reviewer #3

(Remarks to the Author)

The authors perfectly replied to my comments.

REVIEWER COMMENTS

We thank the reviewers and the editor for their thorough review of our manuscript. In response to their comments, we have made substantial changes to the manuscript which improved it. We have addressed each comment in a point-by-point manner below, with our responses indicated in blue text. Changes to the manuscript text have been highlighted in yellow in the revised manuscript file. Line numbers are based on the version with highlighted changes.

Gender choice for the animal study have been included in the title and methods (line 342-343).

Thank you on behalf of all the authors,
Andrea Marzi

Reviewer #1 (Remarks to the Author):

The study introduces a vaccine platform utilizing replicon RNA (repRNA) complexed with LION™ nanocarriers. Key findings include that a single dose of the LION-SUDV encoding SUDV-GP and LION-Combination vaccines provided 100% protection against lethal SUDV challenge in guinea pigs, while the LION-EBOV vaccine encoding EBOV-GP achieved 50% cross-protection. The vaccines elicited robust and multifunctional antibody responses, correlating with decreased viral replication and clinical signs and enhanced survival rates.

Major Points:

The platform-specific differences in immune responses were noted when repRNA/LION vaccines were compared to VSV-based vaccines, with repRNA-LION stimulating a broader functional antibody profile compared to VSV. Generally, the data is well presented and demonstrate the efficacy of this vaccine platforms is associated with an increased phagocytotic activity profile of repRNA vaccine approach versus the VSV platform although both platforms are highly efficacious in this model. In addition, the absence of T-cell data, due to absence of standard method for Guinea is a significant limitation of study from the immunological perspective.

Thank you for the positive evaluation of our data set and acknowledging its limitations particularly the absence of T cell data. This limitation is discussed in the manuscript (lines 249-268).

The novelty of this work mainly resides in the delivery method of the alphavirus RNA replicon, which uses LION as opposed to SUDV-GP virus-replicon particles (VRP). The latter was previously demonstrated to fully protect non-human primates from lethal SUDV challenge (Herbert et al., J Virol 2012). Although technical details of the SUDV GP alphavirus repRNA construct are not clearly described in the methods section, they appear identical to those used by Herbert et al., as the authors later state in the discussion (lines 221 to 225). The main differences between this study and Herbert et al. 2013 lie in the use of guinea pigs versus non-human primates (NHPs) and the delivery method (LION vs VRP). To better highlight the advancement of this work over Herbert et al., the authors should expand demonstrate the advantages of LION over packaged repRNA in VRPs in term of efficacy and/or production. I think this

paper would have greatly benefited from head-to-head comparison LION versus LNP and/or VRPs instead of VSV as we already knew that VRP delivered SUDV-GP repRNA was efficacious in NHPs. This would have demonstrated the potential of LION versus VRP delivery of the repRNA. To the best of knowledge, this type of comparative work has not been published for any repRNA vaccines.

Thank you for this comment. We expanded the methods section to describe the vaccine preparation in much greater detail (lines 302-306). We agree that the comparison of LION vs VRP would be interesting; however, the nature of said comparison is not the focus of this manuscript. The focus is on the efficacy of the LION platform as a single dose SUDV vaccine with using the VSV-SUDV as a positive control vaccine. The LION-SUDV and LION-Combination vaccines had unknown efficacy prior to this study. In the future a comparative study between VRP, LION, and a traditional lipid nanoparticle delivery system may be of interest to our collaborators to determine immunological differences between the delivery methods of the VEEV replicon.

Furthermore, to this day there are no licensed vaccines based on the original VRP even though efficacy has been shown several years ago, seeding doubt about the scalability and quality of GMP product. In contrast, India awarded emergency use approval to the LION-COVID-19 vaccine in 2022 (<https://hdt.bio/hdt-bios-covid-19-vaccine-wins-regulatory-approval-in-india/>). This makes that platform much more valuable and interesting for other re-emerging infectious diseases, like Sudan virus disease. As vaccines based on VSV are in development for all filoviruses and a single-dose vaccine for Ebola virus based on VSV is approved for human use and quite widely used, it made the most sense to use the VSV-SUDV as a control vaccine in our studies (lines 75-90).

Minor points:

Line 171-172 “This result was surprising, given that these guinea pigs did not exhibit detectable cross-reactive antibodies to SUDV GP until 28 DPC.”

This statement is unexpected, as cross-reactive antibodies between Sudan virus and Ebola virus GPs have been described on multiple occasions. I believe the cutoff used for positivity might be responsible for this interpretation. In Figure 3A, B, C, the dotted line represents 'the assay limit of detection,' but the manuscript does not explain where this limit comes from. Additionally, upon reviewing the graphs, it is evident that several animals develop cross-reactive antibodies prior to the challenge, albeit at lower levels.”

Thank you for the insightful comment. Yes, cross-reactive antibodies have been described for EBOV and SUDV GP, including in our own work with NHPs (Marzi et al. Lancet Microbe 2023). The ELISA results here were unexpected since, before challenge (-7 DPC), only 4/10 animals had titers at the limit of detection (1:50) with one having a 1:200 titer. After challenge (6 DPC), we anticipated an increase in EBOV GP-specific antibodies based on cross-reactive epitope boosting; however, we did not see that in this study until the surviving animals fully developed a mature humoral response to the challenge virus on 28 DPC. The statement was intended to emphasize this point as there is a clear difference in the animals presenting the homologous antigen. The limit of detection is the lowest serum dilution tested (1:50). We added this definition to the methods (lines 382-384).

Reviewer #2 (Remarks to the Author):

This paper addresses an important topic, contributing to the response to the recent outbreak of Sudan virus. Having read through, I found several shortfalls which I have listed below.

MAJOR COMMENTS:

(i) The study in guinea pigs was concluded on day 14 post-challenge, whereas routinely day 21 (or even day 28) are widely used for these type of studies. From the clinical data (Figure 1), not all animals seem to be putting on weight and endpoints were reached on day 12. Therefore, the authors need to explain why the study was not scheduled to last longer to capture any animals with late-onset disease signs.

Thank you for the comment. We couldn't agree more with this reviewer and want to point out that all animals that survived challenge were taken out to 28 DPC (line 351). Temperature and weight data were obtained daily until 14 DPC when all animals reached/exceeded baseline weight. We did not obtain weight or temperature data from that day onward as the acute disease phase moved to convalescence and the guinea pigs were healthy. We continued to perform daily health checks from 15 DPC until 28 DPC when the study ended.

(ii) The histology analysis would greatly be aided if viral staining were performed instead of gross H&E changes. Could this staining be conducted? Given the importance of the histology findings, could these images be brought into the main manuscript?

We appreciate this comment. We added the IHC to the histology figure and moved it to the main figures (new figure 3). The results were updated to reflect this addition (lines 138, 143-145).

(iii) Serum samples were collected at day -7, 6 and 28 DPC (line 132). I am not sure why samples weren't collected on the day of challenge, to provide important information on the immune results at the time of challenge which are perhaps the most useful.

Thank you for the comment. We agree that it would be a valuable timepoint for immune response measurement at the time of challenge; however, in our ABSL4 facility it is difficult to perform non-lethal bleeds for guinea pigs at this time, so we were unable to collect this timepoint. It was not justified to include a separate 0 DPC euthanasia group into the study. We compromised and conducted the -7 DPC (one week before challenge) blood draw in ABSL2 to enable antigen-specific ELISA analysis before challenge.

(iv) For the statistical analysis, could the authors please double-check? In Fig 3C (described in line 146) it would appear that the LION-Combination group is significant values are all higher than the control group. Similar in line 147, the LION-EBOV group looks significantly higher but is not mentioned in the text or figure.

We appreciate the attention to detail from this reviewer. After double-checking the statistical analysis, the -7 DPC LION-Combination vs Control is not significant ($p = 0.732$) and neither is the 6 DPC LION-EBOV vs Control ($p = 0.164$), utilizing the Kruskal-Wallis test with Dunn's multiple comparisons.

(v) Line 151-152 mentions about the VSV-SUDV vaccinated animals having evidence of ADCD. At the day - 7 timepoint this was not significant so I can't see how this could be assigned to a vaccine response.

Thank you. This statement was regarding the 6 DPC data where there is a significant difference of ADCD in the VSV-SUDV group. It has been clarified in the text (line 172).

(vi) Line 154-156: In this section, I think that only the LION-SUDV and VSV-SUDV vaccines are directly comparable, as these are the only groups with the same antigen on different backgrounds. Therefore, the argument about VSV giving a more robust profile is questionable.

We appreciate this comment and agree. We amended this sentence to describe the specific differences in functional profiles for LION-SUDV and VSV-SUDV (lines 174-177).

(vii) Lines 170-172: The authors don't seem to correlate the 50% vaccinated with LION-EBOV being protected being identical to the 50% of animals in this group having anti-SUDV antibodies in Fig. 3A. It would be good to clarify if the animals having specific IgG were the ones which were provided protection against virus challenge.

Thank you for this comment, we agree that this would be a very interesting correlation. However, we are unable to make this comparison as the animals that displayed SUDV GP-specific IgG 6 DPC were euthanized for sample collection, and the surviving animals only had their humoral response measured - 7 DPC which were undetectable except for one animal. As such the 50% survival correlate cannot be made to cross-reactive antibody stimulation.

(viii) Line 178-182: With the use of outbred animal species, outliers are often experienced. There is too much emphasis on this single animal and no consideration of effect being due to the nature of the biological systems used.

We appreciate this comment. We added discussion of the possibility that this is a feature of the outbred Hartley guinea pig model being used (line 200-201).

(ix) Line 193: Upon looking at ref 29, the authors appear to be selectively using data which support their work and arguments. The 100-fold increase is only observed in this reference at the day 7 timepoint, and for earlier timepoints viremia is actually lower.

Thank you for pointing this out. We choose to mention these data as the NHP were close to the time of euthanasia when the dramatic difference occurred. As stated in the discussion, the mechanism is unknown, but a 2-log increase is a significant increase of infectious virus at that point of the study. In addition, we added that antigenic difference does not impact protection from MARV and RAVV in NHPs (lines 204-206) supporting the antigenic mismatch not to be a general concern in NHPs but maybe in other animal species.

(x) Lines 235-239: There are pros and cons with all animal model systems. Stating "...makes ferrets more favorable than the guinea pigs" is misleading. Whilst non-adapted virus is preferable, the infectious doses and disease progression kinetics could be argued that guinea pigs are the better system. The authors need to provide balanced arguments. Similarly, the argument of immunological reagents is

invalid. Along with ferret-specific, there are also increasing guinea-pig specific reagents available. For T-cell analysis, there are mAb clones for IFN-gamma commercially available (e.g. MabTech) which can be used for flow cytometry, ELISA and ELISPOT assays.

Thank you for this comment. We know and agree that there are pros and cons to each system and this statement is skewed towards the opinion of the authors and cannot be quantified which model system is better. As such, the statement was removed to limit implicit basis.

(xi) The methods are very brief and miss crucial information. Whilst some details are in the supplementary documentation, key points are still absent.

The source of the non-ATCC cell lines are required, especially to be assured of their authenticity. Apologies for the oversight. ATCC catalogue numbers for all cells used have been added (lines 290, 291, 294, 296).

The differentiation of HL-60 cells to what state is not mentioned, nor assurance of the phenotype of the differentiated cells.

Information has been added (lines 412-413).

The source of the guinea pig adapted virus is needed, and whether this strain is available by others, alongside the passage history during propagation.

Information has been added (lines 299-300).

For the TCID50 method (line 307), key information on the analysis is missing, e.g. CPE assessment/crystal violet staining, controls to standardise outputs, etc.

Information has been added (lines 365-366).

In histopathology (line 313), it just says "prior to staining" with no details on stains.

Hematoxylin & eosin stain has been added (lines 371-372).

For the ELISA (line 322), no information on the secondary antibody is provided.

Requested info has been added (lines 384-385).

The antibody-dependent assays all lack antibody concentration details and controls/standards for interpretation.

Information has been added (lines 397, 399-400, 404, 406-407, 411, 415-416).

In the neutralisation assay (lines 353-359), I am unsure how the readout is on FACS - are cells stained to assess infectivity?

We appreciate the comment. The neutralization assay uses a VSV-SUDV construct encoding GFP which is expressed in infected cells and allows the detection of infected cells by FACS (line 424).

(xii) Lines 285-298: In the animal study design, groups of 4, 5 and 6 animals are used for different analysis. The reasoning for these numbers, e.g. power analysis, is critical.

Thank you for the suggestion and the power analyses have been added to the methods section (lines 339-341, 352-357).

MINOR COMMENTS:

Line 36: comma needed after '2000'. Has been adjusted (line 36).

Line 36: is there a better reference covering the outbreak in Uganda to reference 2?

This outbreak is referenced in the next sentences, reference 3 (lines 36-40).

Line 45: Reference 7 is cited, but key information on the blended vaccine, including a SUDV component, did provide protection. Appreciate that it was just assessed in n=2 animals though.

The monovalent vaccines did not provide protection which is the focus of this statement.

Line 47: add ';' before 'however'. Has been corrected (line 47).

Line 60: space before 'Recently' needs adding. Has been adjusted (line 62).

Line 64: change to "Oil-in water...have been shown..." has been adjusted (line 66).

Line 138: The significantly higher responses are mentioned, but what about the importance of the lower responses? Statement has been added (lines 174-177).

Line 169: "robust" is used to describe responses in contrast to "broad" in line 155. Needs consistency.

Thank you for this comment the statement has been adjusted (line 190).

Lines 202-205: this belongs in the results section.

As the correlation analysis is not a primary figure, we sought to strengthen the argument of the polyfunctional humoral response in the discussion section with these findings.

Lines 205-207: the "Additional effector functions" should be listed.

Thank you for this comment. ADCP and neutralization are listed after this statement (lines 229-231).

Line 206: The "5 DPC" is mentioned, but this doesn't fit with timepoints for the parameters in Fig 3.

Weights were monitored daily, and the correlation analysis being referenced is for the 6 DPC effector function compared to 5 DPC weight change.

Line 233: As ref 6 is cited earlier in sentence would remove from later in sentence.

This has been removed.

Line 263: Mentioned mouse studies being approved by the University of Washington IACUC, but no authors are from this institute.

Dr. Jesse Erasmus is cross-appointed at the University of Washington but his main affiliation is HDT Bio (<https://sites.uw.edu/fullerlabuw/dr-jesse-erasmus/>).

Line 302-303: Using stock TCID₅₀ to quantify RT-PCR outputs is not optimal. Transcript control material for measuring genome copies is more relevant as there is variation in live virus quantification and viral RNA levels.

Thank you for this comment. We agree that there are differences in live virus quantification and viral RNA levels; however, in our opinion, using the stock virus as standard with TCID₅₀ equivalent measurements allows for the RT-qPCR results to be more relevant and comparable to the obtained infectious titers.

Fig 3E and F: Having ADCP and ADNP, respectively, on the x-axis similar to Fig 3D would be beneficial.

We agree with this comment to improve ease of data interpretation and have added the labels to the x-axis (figure 4).

Reviewer #3 (Remarks to the Author):

Donnell et al evaluated the protective efficacy of candidate vaccines based on repRNA vaccine platform. One candidate expresses SUDV-Gulu GP nucleotide sequence (LION SUDV), the second one expresses SUDV and EBOV GPs (LION-Combination) and the third one expresses EBOV GP alone (LION-EBOV). These candidate vaccines were compared to VSV-SUDV vaccine (SUDV Boneface variant) in guinea pig model. The authors observed that a single dose of LION SUDV and LION-Combination induced 100% protection after a challenge with a guinea pig adapted-SUDV Boneface strain three weeks post-immunisation. Interestingly, LION-EBOV vaccination led to 50% protection against a SUDV challenge which means a cross-protection can be observed in some animals. Strong protection induced by LION SUDV and LION-combination correlated with low pathology, low viremia and viral loads in blood, liver and spleen. Protection also correlated with early functional antibody responses.

It is a very elegant and well-designed study. The paper is well written and the figures are quite clear.

Major comments:

Even though this repRNA vaccine platform has been previously described with other sequences, data linked to the LION SUDV and LION-combination vaccine design and formulation would be very relevant before showing immunogenicity and protection data. I mean a few data similar to Fig 1 (Erasmus et al 2020).

We appreciate this comment. We generated the data as suggested and demonstrate efficient antigen expression by immunofluorescence analysis and western blot shown in the new supplemental figure 2. ETS and KH conducted the experiments and were added as coauthors. The results and methods sections have been updated accordingly (lines 101-103, 311-331).

Line 121: As a histopathological analysis was performed by a pathologist, could the authors add a scoring system to their analysis? The scores could be used in the correlation analyses.

We appreciate this comment. The revised version includes the pathology scores as supplemental figure 3 and how the scores were obtained (lines 373-375); however, we did not use the data in the correlation analysis. Tissue sampling always includes a bias because the entire organ is not sampled; therefore, including the pathology scores in the correlation analysis might introduce an unknown variable in the analysis. In future, we will consider including this analysis.

Suppl Fig 2: please add a scale to each picture. The pathology which is described in the text could be shown on the pictures using arrows.

Thank you for this suggestion. This has been added to the new figure 3.

Suppl Fig 4: Could the authors show representative plots for each experimental group ?

We appreciate this comment and included a representative gating strategy for each effector function analysis as new supplemental figure 4 (line 395).

Minor comment:

Please check the word Spearman in the manuscript. Sometimes it is written Spearman.

We appreciate this comment and corrected it to Spearman.

We thank the reviewers and the editor for their thorough review of our manuscript. In response to their comments, we have made the final changes to the manuscript which have been tracked in the revised manuscript file. Line numbers are based on the version with changes.

Previous changes to reviewer's comments are highlighted in yellow.

Thank you on behalf of all the authors,
Andrea Marzi

REVIEWERS' COMMENTS

Reviewer #2 (Remarks to the Author):

The authors have done a great job revising their manuscript, and this second version is a lot easier to read and interpret. I just had a couple of minor comments:

(1) For several of the rebuttals, the information and arguments are acceptable and justified (e.g. feasibilities of working in the ABSL4 laboratory and restrictions with taking bleeds). However, as these haven't all been incorporated into the manuscript, the reader won't be privy to the information and reasonings. Therefore, incorporation of these reasonings into the manuscript would prevent the reader also having the same questions and concerns.

We incorporated the information about the survival blood sample collection from the guinea pigs in the manuscript (lines 372-373). Limitations regarding T cell response analysis are highlighted in lines 271-275.

(2) The figure legends in the text are out of synchronisation with the incorporation of the histology into the main file, so need changing.

Thank you for catching this. We changed the figures labels in the text accordingly (lines 179, 188, 192, 203, 205, 208).

Reviewer #3 (Remarks to the Author):

The authors perfectly replied to my comments.

Thank you, we appreciate the feedback.